# The Effect of Environmental Regulation on Marine Economic Transformation under the Decentralized System: Evidence from Coastal Provinces in China

**Haoran Ge [1,\*], Changbiao Zhong [2], Hanwen Zhang [3] and Dameng Hu [2]**

1. Business School, Zhejiang Wanli University, Ningbo 315100, China
2. Business School, Ningbo University, Ningbo 315100, China
3. Business School, Ningbo City College of Vocational Technology, Ningbo 315100, China
* Correspondence: gehaoran@zwu.edu.cn

**Abstract:** The transformation of the marine economy is a central issue in China's economic sustainability. On the conflicting goals between sustaining a strong marine economy and protecting the environment, this study explored the direct and spillover effects of two types of regional environmental regulation on the marine economic transformation of China's coastal provinces (excluding Hong Kong, Macao and Taiwan) under a decentralized system. By establishing a theoretical framework, using panel data of coastal provinces (cities) in China from 2010 to 2019, and using methods of spatial correlation test and spatial measurement model involved moderator, the results show: (1) The gaps in marine economic transformation were gradually narrowed among these regions, but a significantly negative spatial autocorrelation remained. (2) Incentive-type environmental regulation had a direct effect on marine economic transformation but had a negative effect on the adjacent areas, and the decentralization system could play a positive moderator effect. (3) The investment-type environmental regulation and local marine economic transformation showed a significant "U-shape" relationship, and such regulation had a positive effect on adjacent areas. Decentralization could aggravate the negative effect on the local level but had not yet significantly changed the spillover effect. (4) Presented suggestions for formulating policy, industrial transfer compensation, and regulation decentralization. Hopefully, the findings of this study can shed light on how to improve the efficiency of environmental regulation and realize the sustainable goals of the marine economy.

**Keywords:** marine policy; marine economic sustainability; environmental regulation; decentralization; space spillover



## 1. Introduction

With the prominent advantages of marine resources and marine space, coastal countries and regions are becoming major contributors to world economic growth. In China's coastal provinces (except Taiwan, Hong Kong and Macau), an open economic system dominated by the marine economy has been formed. Since 2019, the proportion of marine economic output in the gross domestic product (GDP) has reached 17.1%.

However, unsustainable development modes and regional GDP championships have increasingly caused serious marine ecological problems due to the publicity of marine resources and the negative externalities of pollution. According to the changing trend of sea areas categorized into the four pollution levels, the proportion of area in level I, the least polluted sea area, only increased from 94% in 2010 to 96.8% in 2020, and the spatial distribution of marine pollution is highly consistent with the marine economic scale in coastal regions of China [1]. At the same time, with the change in international demand structure and the enhancement of trade barriers, the growth rate of the marine economy has reduced from 11% in 2008 to 6.2% in 2019. The marine economic transformation has been becoming more and more urgent in terms of sustainable development.

In its pursuit of sustainable development, China has set up a series of transformation pilot regions, such as "marine economic development demonstration zones", and has imposed environmental regulations as an important means to manage the quality of the marine economy. Although there are more and more ways of environmental regulation in China, investment-type environmental regulation and incentive-type environmental regulation are the most common. The former requires local governments and enterprises to invest in controlling pollutants through laws and standards, and the latter increases the environmental cost and improves the environmental development efficiency of enterprises through market means. The purposes of both regulations are to stimulate enterprises to change traditional ideas of extensive development through appropriate environmental control strategies. The theoretical results that comply with the "Porter hypothesis" are that they can offset the increased costs caused by environmental constraints through innovation and improving resource efficiency. A regional industrial elimination mechanism has been successful in some pilot regions. However, given the frequent events of marine environmental emergencies in recent years, the effect of environmental regulations on marine economic transformation remains unclear.

In addition, the government implements the fiscal and taxation system with centralized politics and a decentralized economy. Local governments are allowed to implement opportunistic and strategic environmental regulations: on the one side, because of the non-exclusive and non-competitive use of marine environments, local governments may pursue the "beggar-thy-neighbor" strategy that can make endogenous pollution exogenous. This will ultimately lead to the downward competition of regional regulations [2]. But on the other hand, local governments may formulate stringent regulatory measures to attract advantageous marine resources and improve economic sustainability [3]. Therefore, under a decentralized system, the purpose and effect of marine environmental regulation will be affected by the relationships among adjacent regions [4].

Previous studies have mainly focused on the direct effect of environmental regulation on the marine economy [5] or the coastal economy [6,7], or on the analysis of the direct stimulating effect of decentralization on the regional economy [8,9]. This paper focuses on researching the moderator mechanism of decentralization systems in the spatial effect of environmental regulations, and it attempts to answer the following questions: (1) Will environmental regulations implemented in China's coastal regions have a nonlinear effect on the transformation of the regional marine economy? (2) Will these regulations have spillover effects on the adjacent regions in the process of policy imitation and competition among regions? (3) Does China's decentralization system affect environmental regulations enacted by local governments significantly? How can we further influence the marine economic transformation? This is more conducive to reflecting the management attitude and effect of regional governments toward the marine environment.

In this study, the second section theoretically summarizes the relationship between decentralization, environmental regulation and marine economic transformation through a literature review. The third chapter introduces the methods of the spatial correlation test and the spatial measurement model with nonlinear terms and moderator terms. The fourth chapter measures the direct effect and spatial spillover effect of marine environmental regulation as well as the moderator effect of decentralization by using the data of 11 coastal provinces and cities in China. In addition, according to the results and problems, the fifth chapter put forward different development strategies and correction measures, which can provide policy recommendations for the sustainability of the marine economy from institutional and managerial perspectives.

## 2. Literature Review and Theoretical Framework

### 2.1. Environmental Regulations and Marine Economic Transformation

There have been several studies on the effect of environmental regulations on economic transformation. The neoclassical economics theory under the hypothesis of invariable technology and demand holds a negative view from the aspects of enterprise cost and industrial performance. The "cost hypothesis" holds that excessive environmental regulations will only increase enterprises' burden of controlling or limiting pollution and slow down the investment of prime capital, which is unfavorable for the transformation of the regional economy. Based on this theory, previous studies had carried out a substantial number of empirical tests, including the analysis of resource-productivity changes in specific industries [10] and specific professions [11]. There are also many comparative research efforts on the relative cost [12] and innovation competitiveness [13] of enterprises, which have verified the inhibitory effect of environmental regulations on economic transformation. Those articles take environmental regulation as an exogenous factor for static analysis and do not consider the changes in enterprises' production behavior under regulation. As such, dynamic research initiatives that include the "Porter hypothesis" have received greater attention. Such studies focus on the transformation choices of enterprises under the impact of environmental regulations. They consider that proper control intensity can reverse the investment in pollution governance in enterprises and lead to innovation, and the increasing marginal cost caused by technical input can be compensated by more efficient productivity [14]. On this basis, many scholars apply relevant theories to the empirical research of specific industries or professions. The results of the analyses in terms of industrial structure [15], labor efficiency [16] and product quality [17] verify the positive role of environmental regulation in economic transformation.

So far, the compensation or substitution effect of environmental regulations in economic transformation remains unclear [18,19]. The reasons are that the environmental constraints in different industries with special technology, resource endowment and market conditions may also be different, and here there are also differences among the transmission mechanisms of different environmental regulations [20]. Hence, the impact of environmental regulations on the marine economy was specially analyzed in several articles.

On the marine economy, previous articles analyzed the impact of environmental regulation on all coastal provinces and cities [21,22], coastal urban agglomerations [23], underdeveloped coastal regions [24], the coastal manufacturing industry [6], coastal tourism [25] or emerging marine enterprises [26]. The research perspectives mainly included economic growth, industrial structure, innovation efficiency, the transfer of polluting industries and green total factor productivity. The results showed that the innovation compensation effect and substitution effect of cost could co-exist, and the role of environmental regulation in economic transformation may be non-linear due to the different levels of innovation [27]. In terms of mechanisms, the marine industry structure, capital investment and marine technology progress were mediating factors impacting the marine economic transformation [5,7].

### 2.2. Decentralization and Regional Economic Development

Research on decentralization mainly focuses on the effects of rights distribution. The existing articles show that decentralization affects regional economies through the level of government competition, the precision of policy implementation and the efficiency of resource allocation [28], of which "fiscal federalism" is the most representative. The theory holds that by decentralizing the right to allocate resources and introducing competition into the market, the central government can construct the Pareto optimization of inter-regional factor distribution, thus improving the overall economic quality [29].

At present, the empirical literature on the economic effect of China's decentralization system is abundant, but the results are different. Zhu Jun and Xu Zhiwei constructed a dynamic stochastic general equilibrium (DSGE) model of multi-level government fiscal policy behaviors and found that the decentralization of policies will promote the economies

of local and surrounding regions [30]. Some scholars believe that China's decentralization system will integrate country administration and regional economic construction into the unified responsibility structure and can improve capital and material resource efficiency [31]. From the perspective of local welfare, Wenqiang Qian also found that it was more conducive to economic sustainability [32]. However, the subsequent refinement articles find that for industries, such as marine-related industries, in which the environmental owners are uncertain, decentralization may increase negative externalities in the process of environmental development and amplify the degree of resource mismatch caused by institutional differences and endowment gaps, thus reducing the development quality of the regional economy [33]. In addition, some scientific studies came to different conclusions, including the inverted U-shaped relationship that is more complex between financial ability and regional economic quality [34].

### 2.3. Decentralization, Environmental Regulation, and Marine Economic Transformation

The different effects of decentralization on the economy are mainly due to the differences in decision-making and strategies of local governments. As the public platform for economic activities, the environment is an important factor in governmental economic management [35]. Early studies have argued that decentralization can accelerate economic growth by improving environmental quality, for example "the voting by foot theory". It suggests that local governments tend to prioritize the needs and services of residents, especially environmental quality, to attract more people and resources. The conclusions are based primarily on the assumption that local governments want to prioritize and ensure the welfare of the population. However, with the enrichment of theoretical hypotheses and empirical verification, scholars have raised many questions about the early results. They believe that complete market and public policy efficiency are easily influenced by the self-interest of local governments, and local governments will inevitably adopt destructive competition around internal economic growth [36]. Empirical studies on typical regions, such as the United States, found that decentralization can indeed stimulate local governments to compete for exploiting the environment (such as the "race to the top" effect and the "not in my backyard" effect) [37]. In terms of the marine economy, the fragmented governance was not conducive to the effect of environmental regulation [38].

According to the existing research, the moderator mechanism of decentralization on the economic effects of marine environmental regulations can be classified into three aspects. Figure 1 shows the mechanism diagram based on existing research, and the explanation of mechanism (a), (b) and (c) are as follows.

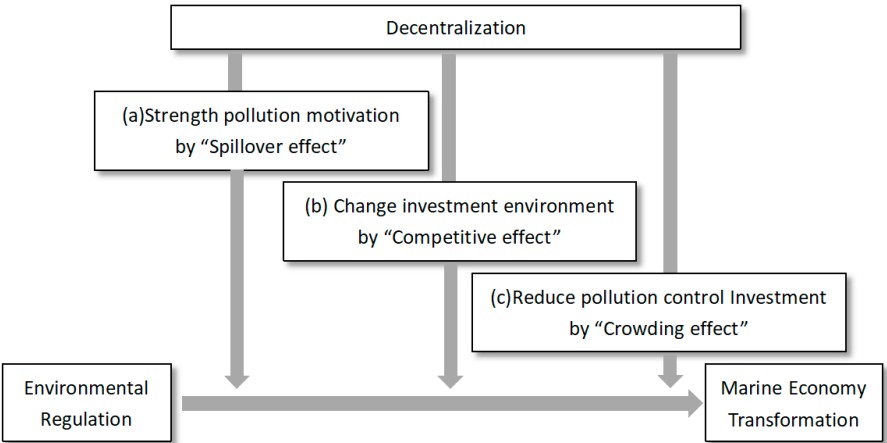

**Figure 1.** The effect mechanism of decentralization, environmental regulation and marine economic transformation. (a), (b) and (c) are, respectively, the economic behavior of local governments under decentralization.

Firstly, as shown in (a) of Figure 1, the decentralization could strengthen the regional pollution motivation, increase the number of polluting enterprises in border areas, and treat the pollutant by "free rider", which may finally affect the normal function of environmental regulation. Specifically, the environmental spillover effect will lead to a mismatch between regional marine economic profit and marine pollution cost. Thus, regional governments tend to establish heavily polluting enterprises in the border area or coastal zone. In a "zero-sum game", this behavior will evolve into the more common phenomenon of the "free rider problem" [39,40]. The deepening of decentralization can further stimulate the regional governments to obtain more benefits with lower environmental costs and deepen the polluting spillover. The negative externalities of pollution will further reduce the transformation expectations of local governments.

Secondly, as shown in (b) of Figure 1, in order to attract more investment than adjacent regions, the regional governments in the decentralization system could change the use of standard marine environments and reduce the effect of environmental regulation. Specifically, the economy of coastal regions has consistently relied on external investment and demand. In order to attract sufficient foreign capital and create more employment opportunities in the short term, local governments will implement environmental regulations with the purpose of "race to the bottom", which will create a regional image of being a "Pollution Haven". This phenomenon also exists in the marine economy. Especially when the benefits of marine environmental improvement cannot compensate for the loss of foreign polluting enterprises, local governments will deregulate the environmental costs of foreign polluting enterprises. It will ultimately affect the efficiency of inter-regional resource allocation and the contribution of marine environmental regulation to the marine economy [41].

Thirdly, as shown in (c) of Figure 1, under the decentralized system, local governments tend to invest in fields with short-term returns. The investment in environmental regulation is squeezed, and the effect is also affected. Specifically, regional environmental management expenditures and other livelihood expenditures are always the opposite. In the case of an expanding financial gap, the allocation of financial funds will be tilted toward activities that generate higher short-term income, such as productive public services [42,43]. The reason is that the marine environmental governance system needs a long-term capital investment. Under the decentralized system, in which greater attention is given to economic income and social welfare in the evaluation system for the local officials' promotion, they tend to be opportunistic and ignore the roles of environmental governance on marine economic transformation [44,45].

Based on the above analysis, there are few studies on the marine economic transformation, but some scholars believe that the "fragmentation" of environmental management under decentralization makes the decentralization system a decisive factor in the resource development of the micro-subject, which is more obvious in the marine economy [46]. This is because the non-division of marine environments and the spatial fluidity of marine resources are more conducive for local governments to adopt environmental hitchhiking measures. Therefore, the change in interest distribution and relationships among regions [47] will cause the transformational orientation of the marine industry to deviate [48].

Compared with the above studies, the innovation of this paper is mainly in the following aspects:

1. At present, academic studies mainly investigate the linear effect of environmental regulations on economic development from a macroscopic perspective. Many of these investigations ignore the possibility of non-linear effects, which substantiate the "environmental Kuznets curve". Based on the existing research, this paper introduces quadratic terms into the spatial economic model, and further distinguishes the direct effect of two types of regulations on marine economic transformation.

2. Existing research emphasizes the fiscal expenditure effect of central government. However, the marine environment of a public pond determines that the spatial spillover effect of regional environmental regulations will also affect the development mode of the local marine economy. As such, this paper introduces the spatial econometric method to quantitatively discuss the regional spillover effect of environmental regulations.

3. The further introduction of decentralization will affect the competition of environmental governance among adjacent governments, and the effects of environmental regulation are also affected. Hence, this paper focuses on researching the moderator mechanism of decentralization systems in the spatial effect of environmental regulations.

Figure 2 illustrates the research framework, in which the box represents the core variable of the article. (a), (b) and (c) represent the direct effect of environmental regulation on local marine economic transformation, the spillover effect of environmental regulation on adjacent regions' marine economic transformation and the moderator effect of decentralization on both effects of environmental regulation.

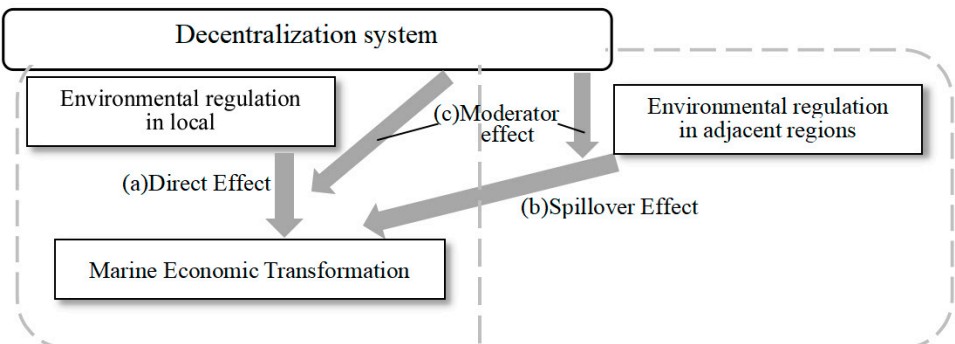

**Figure 2.** The research framework. This study mainly researches the direct effect and spillover effect of marine environmental regulation as well as the moderator effect of decentralization systems. (a) represents the direct effect of environmental regulation on local marine economic transformation. (b) represents the spillover effect of environmental regulation on adjacent regions' marine economic transformation. (c) represents the moderator effect of decentralization on both effects of environmental regulation.

Through theoretical and empirical analyses of these issues, this paper seeks to provide policy recommendations for the sustainable development of the marine economy from institutional and managerial perspectives.

## 3. Materials and Methods

### 3.1. Regions for This Study

China is the second-largest economy in the world and continues to be the largest contributor to world economic growth. This achievement depends not only on abundant resource endowments, but also on the economic development mode of constant transformation. Among which, the coastal areas have developed into the most representative areas with a long coastline of 18,000 km, a jurisdictional sea area of 3 million square kilometers and enjoy many flexible policies. The research area of this paper includes 11 coastal provinces (autonomous regions and municipalities) in China. Due to the large amount of missing data in Taiwan, Hong Kong and Macau, they were not included in the research scope. The study area is shown in Figure 3.

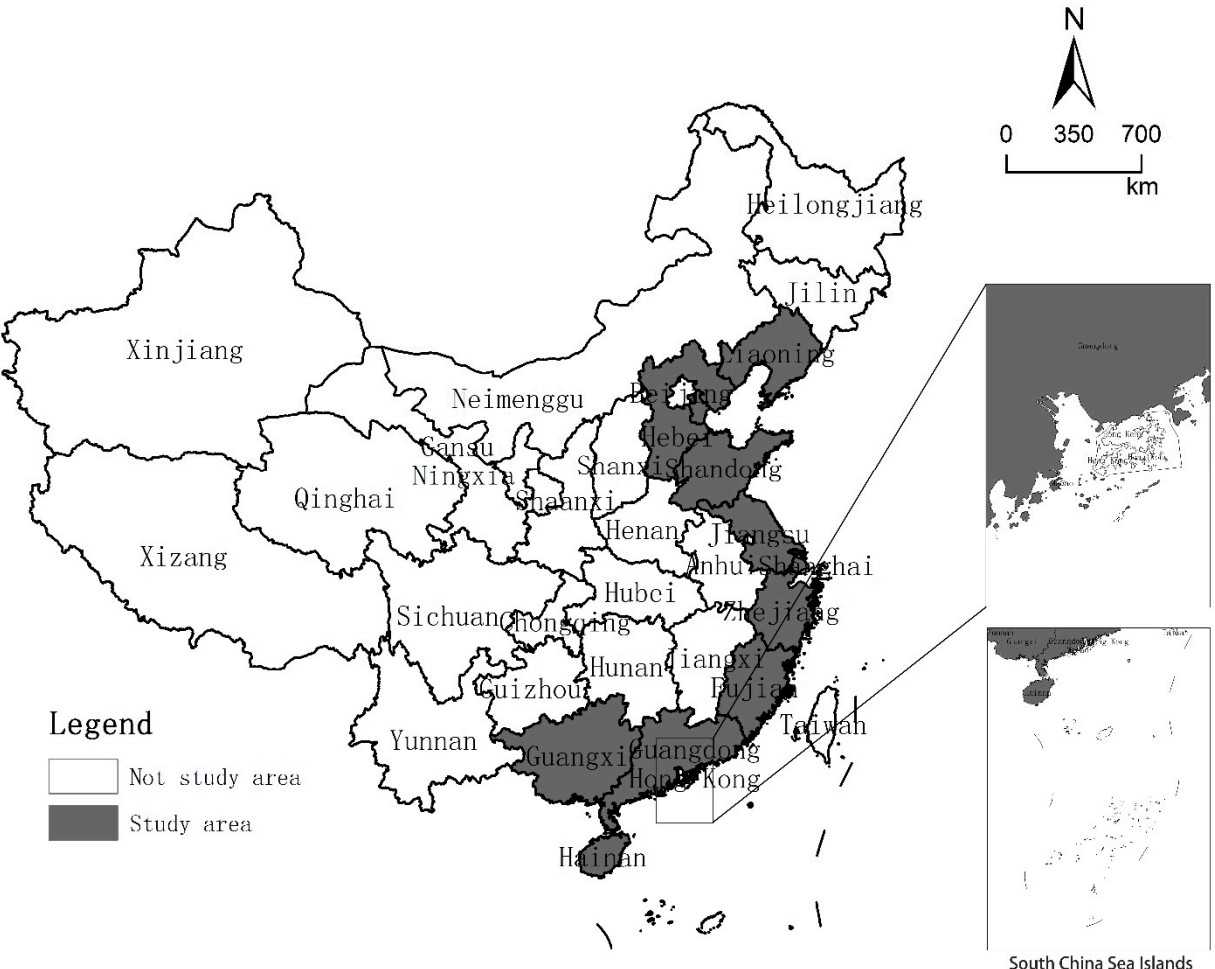

**Figure 3.** Geographical location map of provinces (or cities) in China. The map was carefully drawn by the author using ArcGIS.

### 3.2. Research Methods

3.2.1. Spatial Autocorrelation Test

As can be seen from the theoretical framework in the previous studies, regional environmental regulation may be affected by the imitative competition between governments under the motivation of decentralization, and there are spatial spillover effects of environmental regulation among regions. The spatial autocorrelation test of the main variables (marine economic transformation and environmental regulation) should be carried out before establishing the panel data model. The formula is:

$$\text{Global Moran's I} = \frac{\sum_{i=1}^{n}\sum_{j=1}^{n} W_{ij}(X_i - \overline{X})(X_i - \overline{X})}{S^2 \sum_{i=1}^{n}\sum_{j=1}^{n} W_{ij}} \quad (1)$$

$$S^2 = \frac{1}{n}\sum_{i=1}^{n}(X_i - \overline{X})^2 \quad (2)$$

$$\overline{X} = \frac{1}{n}\sum_{i=1}^{n} X_i \quad (3)$$

where $X_i$ and $X_j$ are the observed values of city i and city j, and $W_{ij}$ is the spatial weighting matrix. The value of Moran's I is between −1 and 1. A value higher than 0 significantly indicates that there is a positive correlation between sample observations in the spatial distribution, which means that similar observation values tend to be concentrated in

adjacent space. A value that is significantly less than 0 indicates that there is a negative correlation in the spatial distribution of the observed values; that is, the observed values with large gaps tend to be concentrated in adjacent space.

As the spatial weight matrix directly determines the possibility of interaction between regions, there are many defining methods, such as adjacency, geographical inverse distance and economic distance, which are combined with the research characteristics in different studies. Given the imitative competition of the adjacent regions and the pollution transmission characteristics of distance attenuation, the spatial weight matrix, which considers adjacency and distance, is selected to reflect the interactive effects of environmental governance between different provinces:

$$W_{ij} = \begin{cases} 1 \text{ if } d_{ij} \leq d_{ik} \\ 0 \text{ if } d_{ij} > d_{ik} \end{cases} \tag{4}$$

where $d_{ij}$ is the spatial distance from province i to j, and $d_{ik}$ is the distance threshold, k is defined as 4 in light of the chained distribution characteristics of China's coastal provinces. Therefore, $d_{ik}$ represents the distance from province i to the 4th nearest province.

### 3.2.2. Construction of the Spatial Econometric Model

Considering that there may be a phase effect, which includes the "Porter hypothesis" or "cost hypothesis" of marine economic transformation under the intervention of government environmental policy. Therefore, when judging the comprehensive effect of environmental regulation, this paper introduces linear terms and quadratic terms. Considering that economic development may depend on an inertial path, this paper also introduces the lag term of the Marine Economic Transformation Index to reduce the deviation caused by time series. In order to eliminate the multicollinearity and heteroscedasticity problems in the model as far as possible, each variable has been processed with a logarithm, and the basic model formula is:

$$\text{LnUPG}_{it} = \alpha_0 + \rho \text{LnUPG}_{it-1} + \alpha_1 \text{LnER}_{it} + \alpha_2 [\text{LnER}_{it}]^2 + \alpha_3 \text{LnFD}_{it} + \alpha_4 X_{it} + \mu_i + \varphi_t + \epsilon_{it} \tag{5}$$

On this basis, the space-lag terms should be introduced to reflect the spatial spillover effect of core variables in more detail, as well as to avoid the endogenous problems caused by the spatial effect on the basic model. To date, the spatial lag model, spatial error model, and spatial Durbin model are the most popular spatial economic models, and the first two consider the spatial effects of independent variables and error items, respectively, while the spatial Durbin model unifies both of them. Based on research objectives, this paper introduces the space-lag terms of "marine economic transformation" and "environmental regulation" and constructs the dynamic space Durbin model.

$$\text{LnUPG}_{it} = \alpha_0 + \rho \text{LnUPG}_{it-1} + \beta_1 (W \times \text{LnUPG}_{it}) + \alpha_1 \text{LnER}_{it} + \alpha_2 [\text{LnER}_{it}]^2 + \beta_2 (W \times \text{ER}_{it}) + \alpha_3 \text{LnFD}_{it} + \alpha_4 X_{it} + \mu_i + \varphi_t + \epsilon_{it} \tag{6}$$

where $\text{UPG}_{it}$ is the marine economic transformation index of the coastal region i in year j; $\text{ER}_{it}$ is the environmental regulation intensity adopted by the region i in year t; $\text{FD}_{it}$ is the decentralization level of region i in year j; $X_{it}$ is the control variable; and W is the spatial weight matrix, same as the above formula. The $\mu_i$ and $\varphi_t$ are used as space and time fixed items to control sample space and time differences; $\varepsilon_{it}$ is an error item(the same below). In the model, $\alpha_1$, $\alpha_2$, and $\beta_2$ are the estimation parameters that should be focused on. These reflect the local effect and spatial spillover effect of environmental regulations on marine economic transformation.

On this basis, to analyze the moderator effect of the decentralization system on the direct effect, the model (7) is created mainly by introducing the interaction terms, including environmental regulation and decentralization; $\theta_1$ and $\theta_2$ should also be focused on.

$$
\begin{aligned}
\text{LnUPG}_{it} = \quad & \alpha_0 + \rho\text{LnUPG}_{it-1} + \beta_1(W \times \text{LnUPG}_{it}) + \alpha_1\text{LnER}_{it} + \alpha_2[\text{LnER}_{it}]^2 \\
& + \beta_2(W \times \text{ER}_{it}) + \alpha_1\text{LnER}_{it} + \theta_1\text{LnFD}_{it} \times \text{LnER}_{it} \\
& + \theta_2(\text{LnFD}_{it} \times W \times \text{LnER}_{it}) + \alpha_4 X_{it} + \mu_i + \varphi_t + \epsilon_{it}
\end{aligned}
\tag{7}
$$

### 3.3. Index and Data Sources

The impact of environmental regulation on the economy of coastal areas is comprehensive. In order to reflect the impact of marine environmental regulation on the marine economy accurately and avoid the estimation error caused by improper data, this paper refers to previous research. The core variables in this paper are calculated from the relevant data in the marine environment and marine economy.

### 3.3.1. Marine Economic Transformation Variable

The measurement of economic transformation in previous studies mainly focused on the industrial advancement level or rationalization level, while the contradiction of our study is whether environmental regulations improve the efficiency of marine resource utilization by changing the department structure. Thus, this paper uses the structure similarity coefficient method, which represents the industrial advancement level, to reflect the marine economic transformation of each province. In terms of calculation, we build the spatial vector of marine industrial structure $X_0 = (x_{1.0}, x_{2.0}, x_{3.0})$, and measure the intersection angles that were named $\theta_1$, $\theta_2$, and $\theta_3$ separately with unit vector $X_1 = (1,0,0)$, $X_2 = (0,1,0)$, $X_3 = (0,0,1)$. The greater of $\theta$ indicates that the corresponding industry share is smaller. The calculation formula is:

$$
\theta_j = \arccos \frac{\sum_{i=1}^{3}\left(x_{i,j} \cdot x_{i,0}\right)}{\left(\sum_{i=1}^{3} x_{i,j}^2\right)^{1/2}\left(\sum_{i=1}^{3} x_{i,0}^2\right)^{1/2}}
\tag{8}
$$

In the formula, $X_0$ is the three-dimensional vector based on the percentages of three industries in GDP. $x_{i,j}$ is the i-th component in unit vector $X_j$. The advancement level of the regional marine economy according to the weights of each industry is then calculated. This paper holds that the marine economic transformation follows the ascending path of regional economies. According to previous studies, the industries of agriculture, manufacturing, and services are given weights of 1, 2, and 3, respectively [49]. The formula for marine economic transformation is:

$$
\text{UPG} = \sum_{k=1}^{3} \sum_{j=1}^{k} \theta_j
\tag{9}
$$

### 3.3.2. Environmental Regulation Variables

Given that there are significant differences in the types of participating subjects and control objectives involved in different environmental regulations, previous research efforts have measured the intensity of environmental regulations from various aspects, among which the indexes of government input and regulation effect are the most popular [50]. Considering that the research purpose is to identify the influence of government management on the marine economy, the input indicators are more appropriate. This paper uses incentive-type environmental regulation and investment-type environmental regulation to respectively represent indirect adjustment intensity and direct input intensity [51].

The incentive-type environmental regulation (abbreviated as ER1) means that the government introduces taxes and other means of market management to internalize the cost of external pollution by enterprises, which may force enterprises to engage in energy conservation and emission reduction. For the marine economy, this mainly includes a tradeable sewage licensing system, direct pollution fees, pollution subsidies and others. In this paper, the fee for a unit of sea area was calculated using the measurement data. The investment-type environmental regulation (abbreviated as ER2) refers to governmental input of pollution disposal, which mainly includes capital input. This paper selects the pollution management investment of the unit ocean output as the agent variable. The formulas are:

$$ER1_{it} = \frac{CSU_{it}}{SA_{it}} \tag{10}$$

$$ER2_{it} = \frac{IPC_{it} \times P_{it}}{MAR_{it}} \tag{11}$$

where $ER1_{it}$ and $ER2_{it}$ are incentive-type environmental regulation and investment-type environmental regulation of area i in year t. SA and CSU are the marine areas of right confirmation and fee of sea utilization, respectively. IPC, P and MAR are governmental industrial pollution control investments, ratio of marine economy output in GDP and marine industry output, respectively.

### 3.3.3. Decentralization Variable

Scholars mainly choose regional fiscal data as the proxy data of decentralization intensity because fiscal decentralization is the basis of other rights divisions, including the administrative right of environmental management [8,52], and the data of local fiscal autonomous revenue and expenditure are the most representative [53]. However, the direct fiscal data cannot accurately reflect the regional power in decentralization because not all the fiscal revenue can be expended [54]. Therefore, this paper chose a degree of fiscal freedom that reflects the government's supply capacity of fiscal revenue relative to fiscal demand as a proxy indicator.

$$FD_{it} = \frac{FR_{it}}{FE_{it}} \tag{12}$$

where $FR_{it}$ and $FE_{it}$, respectively, indicate the regional budget's fiscal income and fiscal expenditure.

### 3.3.4. Control Variables

In order to reduce the estimate bias of the core variables, the paper mainly selects the following control variables: (1) Opening level ($FDI_{it}$): As an important method for coastal provinces to develop regional economies, foreign capitals will change the transformation capacity of regional economies in the integration of international industrial chains. This paper adopts the contribution rate of foreign investment in GDP to reflect the opening level. (2) Technology innovation level ($RD_{it}$): The fundamental path of economic transformation is to increase the efficiency of capital output through innovation. This paper introduces the level of marine science and technology innovation as the proxy data. (3) Infrastructure level ($INF_{it}$): Perfect infrastructure is conducive to the agglomeration of high-quality capital and the scale effect, and enterprises in regions with better infrastructure can release more capital and space for innovation and transformation. (4) Contribution of the marine economy ($EC_{it}$): The proportion of marine economic output in GDP determines its dominance in regional economies. A greater proportion means that there is a higher likelihood that the transformation of the marine economy will be valued by the government. (5) Financial supply capacity ($FS_{it}$): Financial support ability is an important consideration for regional enterprises when deciding whether to transform. A good financial environment can afford the consumption of sinking costs during the process of transformation. (6) Resource endowment level ($RE_{it}$): The marine industry is resource-dependent, so resource endowment

can inevitably affect the production cost and transformation choice of marine enterprises. Table 1 shows the control variables and measurement indicators.

**Table 1.** Control Variables and Interpretation.

| Variables | Symbol | Measurement Indicators | Data Source |
|---|---|---|---|
| Opening Level | FDI | Foreign investment/Gross Domestic Product | China Urban Statistical Yearbook, China Statistical Yearbook |
| Technology Innovation Level | RD | Marine Science and Technology Innovation Investment/Gross Marine Product | China Urban Statistical Yearbook, China Ocean Statistical Yearbook |
| Infrastructure Level | INF | Highway Total Mileage/Area | China Statistical Yearbook |
| Contribution of The Marine Economy | EC | Marine Economic Output Value/Gross Domestic Product | China Ocean Statistical Yearbook, China Statistical Yearbook |
| Financial Supply Capacity | FS | Deposit loan Balance and Insurance Income/Marine Economic Output Value | China Financial Statistical Yearbook, China Statistical Yearbook |
| Resource Endowment Level | RE | Shoreline length/Year-end Total Population | Provincial Statistical Yearbook, China Statistical Yearbook |

### 3.3.5. Data Validation and Source

Due to the COVID-19 pandemic, most of China's marine environmental regulations have been suspended in 2020, and the continuity of the marine economic transformation in adjacent years has been affected. Therefore, the data from 2010 to 2019 are used. In order to reduce the influence of data heteroscedasticity on estimated results, all indexes are processed logarithmically, and the descriptive statistics are shown in Table 2.

**Table 2.** Descriptive statistics of morphological variables.

| | Min Value | Max Value | Average Value | Variance |
|---|---|---|---|---|
| Ln(UPG) | 6.208 | 7.338 | 6.782 | 0.240 |
| Ln(ER1) | 2.018 | 6.072 | 4.222 | 0.891 |
| $[Ln(ER1)]^2$ | 4.074 | 36.869 | 18.615 | 7.295 |
| Ln(ER2) | 0.127 | 3.251 | 0.836 | 0.687 |
| $[Ln(ER2)]^2$ | 0.016 | 10.571 | 1.172 | 1.859 |
| Ln(FD) | 0.312 | 0.668 | 0.518 | 0.103 |
| Ln(FDI) | 2.007 | 4.850 | 3.741 | 0.669 |
| Ln(RD) | 3.522 | 6.704 | 5.287 | 0.746 |
| Ln(FS) | 11.057 | 12.880 | 12.009 | 0.523 |
| Ln(INF) | 1.396 | 3.251 | 2.618 | 0.427 |
| Ln(RE) | 0.064 | 0.615 | 0.261 | 0.157 |
| Ln(EC) | 1.649 | 3.651 | 2.741 | 0.576 |

This paper also tests the multiple collinearity, which can lead to biased estimation results: Firstly, the correlation coefficients among the variables were tested (Table 3). The average variance inflation factor of the basic model is 3.58, less than 5. The results show that there is no serious multiple collinearity among the overall and individual variables, so the above data can be used.

**Table 3.** Correlation among the morphological variables.

| | (1) | (2) | (3) | (4) | (5) | (6) | (7) | (8) | (9) | (10) | (11) |
|---|---|---|---|---|---|---|---|---|---|---|---|
| Ln(UPG) | 1 | | | | | | | | | | |
| Ln(ER1) | −0.249 *** | 1 | | | | | | | | | |
| [Ln(ER1)]$^2$ | −0.255 *** | 0.890 *** | 1 | | | | | | | | |
| Ln(ER2) | −0.352 *** | 0.225 | 0.198 | 1 | | | | | | | |
| [Ln(ER2)]$^2$ | −0.214 *** | 0.196 | 0.080 | 0.840 *** | 1 | | | | | | |
| Ln(FD) | 0.393 *** | −0.233 *** | −0.382 *** | −0.359 *** | −0.314 ** | 1 | | | | | |
| Ln(FDI) | 0.239 *** | −0.346 *** | −0.368 *** | −0.27 | −0.199 | 0.672 *** | 1 | | | | |
| Ln(RD) | 0.213 * | −0.220 * | −0.267 *** | −0.235 | −0.146 | 0.577 ** | 0.428 * | 1 | | | |
| Ln(FS) | −0.072 * | 0.353 *** | 0.375 *** | −0.063 | −0.070 | −0.136 | −0.307 *** | 0.049 | 1 | | |
| Ln(INF) | −0.216 *** | 0.354 *** | 0.311 *** | 0.367 *** | 0.220 | −0.104 ** | −0.278 *** | 0.052 | 0.084 | 1 | |
| Ln(RE) | −0.221 *** | 0.009 | −0.031 | 0.229 | 0.142 | 0.012 | −0.090 | −0.147 | −0.283 *** | −0.118 | 1 |
| Ln(EC) | 0.270 *** | −0.478 *** | −0.485 *** | −0.353 | −0.257 | 0.229 *** | 0.389 ** | 0.208 ** | −0.413 ** | −0.268 *** | 0.026 |

Note: The symbols of *, **, *** in the table show that the results were significant at 90%, 95% and 99% levels, respectively.

Relevant data in this paper were collected from the "China Statistical Yearbook", "China Urban Statistical Yearbook", "China Ocean Statistical Yearbook", "China Environmental Statistical Yearbook", "China Financial Statistical Yearbook" and "China Population Employment Yearbook ", as well as statistical yearbooks of coastal provinces published from years of 2011 to 2020. The missing data on fee of sea area utilization are supplied in two aspects: firstly, it is obtained by referring to the bulletin on the use and management of sea areas issued by the State Oceanic Administration or coastal provinces (cities); secondly, it is obtained by referring to the bulletin on the listing and transfer of sea areas published on the official website of the provincial Ocean and Fishery Department.

## 4. Results

### 4.1. Descriptive Analysis of Marine Economic Transformation

The overall evolution trend of China's marine economic transformation can be derived from the marine economic transformation variable of coastal regions, limitation of paper space, the representative results of 2010, 2013, 2015, 2017 and 2019 are shown in Table 4.

**Table 4.** Value of marine economic transformation among 2010 to 2019.

| | 2010 | 2013 | 2015 | 2017 | 2019 |
|---|---|---|---|---|---|
| Tianjin | 6.749 | 6.724 | 6.704 | 6.783 | 6.914 |
| Hebei | 6.934 | 6.824 | 6.723 | 6.847 | 6.951 |
| Liaoning | 6.469 | 6.472 | 6.511 | 6.649 | 6.787 |
| Shanghai | 7.099 | 7.191 | 7.280 | 7.309 | 7.338 |
| Jiangsu | 6.955 | 6.892 | 6.818 | 6.711 | 6.703 |
| Zhejiang | 6.893 | 6.81 | 6.770 | 6.857 | 6.961 |
| Shandong | 6.759 | 6.754 | 6.753 | 6.856 | 6.959 |
| Fujian | 6.651 | 6.675 | 6.722 | 6.784 | 6.845 |
| Guangdong | 7.044 | 7.018 | 7.002 | 7.069 | 7.116 |
| Guangxi | 6.399 | 6.286 | 6.238 | 6.388 | 6.517 |
| Hainan | 6.580 | 6.790 | 6.758 | 6.732 | 6.706 |
| Maximum | 7.099 | 7.263 | 7.280 | 7.316 | 7.338 |
| Minimum | 6.399 | 6.281 | 6.238 | 6.367 | 6.517 |
| Range | 0.700 | 0.982 | 1.042 | 0.949 | 0.821 |
| Average | 6.776 | 6.768 | 6.753 | 6.821 | 6.891 |
| Standard Deviation | 0.232 | 0.288 | 0.306 | 0.277 | 0.252 |

According to the average values of marine economic transformation in five representative years, there was a trend of decrease first and then increase, and the change range was obviously lower than the average value. This showed that the development of China's marine industry had not divorced from the traditional path of "Pollution first and treatment later". The range and standard deviation of regional values showed a trend of first increasing and then decreasing, which indicated that the transformation level of the marine economy among coastal regions gradually became consistent.

In order to analyze the relative change characteristics of every coastal province more accurately, the method of natural breaks in ArcGIS10.2 software was used to classify the transformation values in the years of 2010, 2015 and 2019. Figure 4 illustrates the spatial classification results.

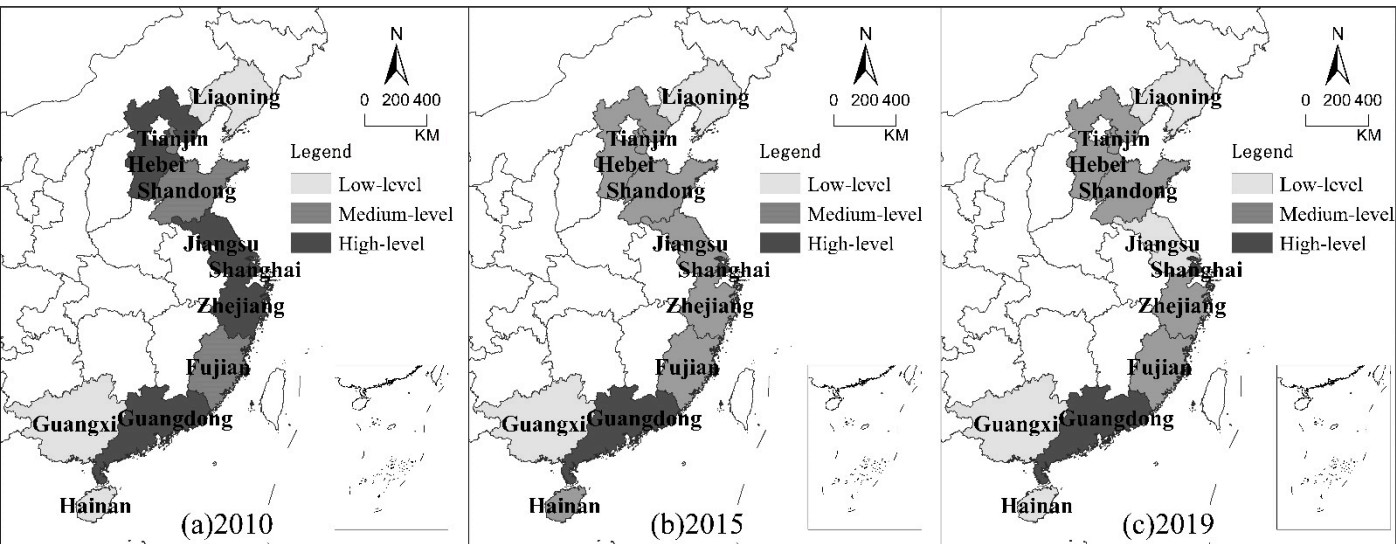

**Figure 4.** Spatial classification of coastal regional marine economic transformation (**a–c**) are the results of 2010, 2015 and 2019, respectively. The map only shows some coastal areas of China.

The classification distribution of regional marine economic transformation was gradually concentrated from the high level to the middle level, and there were significant differences in the variation tendency of each province. The number of high-level regions decreased from five in 2010 (including Hebei, Jiangsu, Shanghai, Zhejiang and Guangdong) to two in 2015 (including Shanghai and Guangdong) and remained unchanged in 2019. The median-level regions increased to seven in 2015 (including Tianjin, Hebei, Shandong, Jiangsu, Zhejiang, Fujian and Hainan) from three in 2010 (including Tianjin, Shandong and Fujian) because Hainan and Jiangsu fell to low levels in 2019. Three regions were at the low level in 2010 (including Liaoning, Guangxi, and Hainan) and fell to two in 2015 (including Liaoning and Guangxi). The level in both Jiangsu and Hainan declined in 2019, and the number in the low-level areas increased to four. The classification results also showed that the transformation effects of regions are more consistent.

According to the spatial layout, the advantages of some leading regions in regional marine economic transformation were more obvious, and there was core-edge differentiation in typical regions. For Shanghai and Guangdong, with a relatively complete industrial system and advanced ocean-related elements, the development of marine resources became more sustainable. However, combined with the conclusions of classification distribution, it showed that these regions had not played the positive spillover effect in overall China's marine economy. Although the surrounding regions such as Jiangsu, Zhejiang and Hainan had volume advantages that could provide more ore, fish and environment, the transformation pressure from adjacent core regions was still serious because of the industrial division. The marine environmental regulations inevitably have spillover effects.

*4.2. Spatial Correlation Testing*

Based on the calculated values of marine economic transformation, incentive-type environmental regulation and investment-type environmental regulation using the software of Geoda, the Global Moran's I indexes from 2010 to 2019 were calculated by setting the spatial relation matrix. Combined with the results of spatial distribution, we can determine the spatial relationship of core variables. The results listed in Table 5 show that there

were significant spatial autocorrelation phenomena in all three variables, among which the Moran's I indexes of the Marine Economic Transformation were negative. Moreover, the index values and significance level were much higher in the last two years. The results indicated that there was a more obvious negative correlation among adjacent regions. Combined with the results of spatial classification, it showed that the "siphon effect" of the core areas was higher than the "radiation effect" in the process of marine economic transformation. The global Moran's I indexes of incentive-type environmental regulation were also significantly negative after 2011. Combined with the spatial distribution characteristics of Marine Economic Transformation, we can explain that the central-peripheral division in the marine industry was spontaneously formed between the core area and adjacent areas. The core area would upgrade the marine industry by improving the environmental use standards. In order to undertake the industrial spillovers of the core area, the developing areas loosened the market entry restrictions on inefficient marine enterprises. The results are consistent with the "Pollution Haven Hypothesis". There were significant spatial positive correlation phenomena in the investment-type environment regulation. For this reason, unlike incentive-type environmental regulation, local investment-type environmental regulation was the important achievement assessment standards from the superior government. In order to avoid punishment caused by insufficient investment, there was obvious comparison and imitation among adjacent regions.

**Table 5.** Values of marine economic transformation among 2010 to 2019.

| Year | Marine Economic Transformation | | Incentive-Type Environmental Regulation | | Investment-Type Environmental Regulation | |
|---|---|---|---|---|---|---|
| | Moran's I | Z-Score | Moran's I | Z-Score | Moran's I | Z-Score |
| 2019 | −0.3649 *** | −2.9243 | −0.551 * | −1.1647 | 0.4526 * | 1.8642 |
| 2018 | −0.3306 *** | −3.5564 | −0.326 ** | −1.1829 | 0.3212 ** | 1.8989 |
| 2017 | −0.2084 * | −1.8268 | −0.571 * | −1.7861 | −0.0143 | −0.1642 |
| 2016 | −0.2554 * | −1.9376 | −0.6273 ** | −1.8828 | 0.2982 *** | 2.8753 |
| 2015 | −0.2363 * | −1.8843 | −0.3495 ** | −2.0943 | 0.3209 *** | 2.9662 |
| 2014 | −0.2797 ** | −1.9728 | −0.0850 *** | −2.5977 | 0.3094 *** | 2.9207 |
| 2013 | −0.3094 ** | −2.0079 | −0.201 *** | −3.7237 | 0.4404 *** | 3.6943 |
| 2012 | −0.2492 * | −1.8842 | −0.1508 | −0.2356 | 0.4722 *** | 3.5575 |
| 2011 | −0.3120 ** | −1.9883 | −0.0583 | −0.1459 | 0.4907 *** | 3.5628 |
| 2010 | −0.3454 ** | −2.0407 | −0.1457 | −0.1608 | 0.4525 * | 1.8642 |

Note: The symbols of *, **, *** in the table show that the results were significant at 90%, 95% and 99% level, respectively.

### 4.3. Spatial Measurement Analysis of Environmental Regulation and Marine Economic Transformation

Based on the spatial autocorrelation test of the core explanatory variables and the interpreted variables, as well as the objective of this study, it is more appropriate to use the spatial econometric model in this paper. The existing spatial econometric models mainly include the Spatial Lag Model (SAR), the Spatial Error Model (SEM) and the Spatial Dubin Model (SDM). A series of tests were conducted to determine the most suitable spatial econometric model, and the results are shown in Table 6. The original assumption of the Lagrange multiplier test (LM test) is that there are no spatial autoregressive and independent variable correlation terms. The results of the LM test showed that both SEM and SAR are applicable. Therefore, we should reject the original assumption, and the SDM should be considered. The results of the LR and Wald tests are also significant, showing that the SDM could not degenerate into SEM or SAR, which indicated that SDM model can well describe the spatial correlation of marine economic transformation.

**Table 6.** The selection test of spatial econometric model.

| Test Method | Index | Value | Test Method | Index | Value |
|---|---|---|---|---|---|
| LM Test | LM_Error | 187.672 *** | LR Test | LR_SAR | 33.920 *** |
| | RLM_Error | 37.536 *** | | LR_SEM | 20.980 *** |
| | LM_Lag | 274.336 *** | Wald Test | Wald_SAR | 35.380 *** |
| | RLM_Lag | 154.200 *** | | Wald_SEM | 20.960 *** |

Note: The symbols of *** in the table show that the results were significant at 99% level.

In order to test whether there were estimation errors caused by individual differences or time differences in panel data regression, it is necessary to judge the fixed and stochastic effects of the model before making a spatial metering analysis. This paper used the Hausman test to verify the space Durbin model. The Hausman statistics indexes of the basic model introduced, ER1 and ER2, respectively, were 39.3257 and 20.2248, and the significance test of the *p* value rejected the original assumption of the stochastic effect, which explained that the time-space double fixed space Durbin model should be used. Using the software of stata, Table 7 shows the regression results.

**Table 7.** The spatial econometric results of environmental regulation for Marine Economic Transformation.

| Regulation Type | Incentive-Type | | | | Investment-Type | | | |
|---|---|---|---|---|---|---|---|---|
| Result | (1) | (2) | (3) | (4) | (5) | (6) | (7) | (8) |
| Ln(ER1) | 0.0481 *** (3.34) | 0.0397 *** (2.82) | 0.0268 (0.35) | 0.0363 *** (2.57) | | | | |
| [Ln(ER1)]$^2$ | | | 0.0019 (0.21) | −0.0008 (−0.11) | | | | |
| Ln(ER2) | | | | | 0.0176 (1.16) | 0.0404 *** (2.78) | −0.2673 *** (−4.28) | −0.2832 *** (−5.10) |
| [Ln(ER2)]$^2$ | | | | | | | 0.0192 *** (4.13) | 0.0259 *** (5.07) |
| W × Ln(ER1) | | −0.0232 *** (−2.56) | | −0.0247 *** (−2.63) | | | | |
| W × Ln(ER2) | | | | | | 0.0252 *** (2.81) | | 0.0203 *** (2.75) |
| Ln(FD) | −0.0452 (−0.27) | −0.0097 (−0.54) | −0.0417 (−0.25) | −0.0639 (−0.38) | −0.0206 (−0.11) | −0.1074 (−0.69) | −0.1326 (−0.82) | −0.2389 (−1.58) |
| Ln(FDI) | −0.0245 (−0.79) | −0.0127 (−0.39) | −0.0246 (−0.79) | −0.0208 (−0.68) | −0.0456 (−1.36) | −0.0134 (−0.44) | −0.0416 (−1.43) | −0.0392 (−1.35) |
| Ln(RD) | 0.0618 *** (−2.85) | 0.0704 *** (3.14) | 0.0626 *** (2.98) | 0.0713 *** (3.58) | 0.0293 (0.30) | 0.0704 *** (3.45) | 0.0587 *** (2.84) | 0.0654*** (3.09) |
| Ln(FS) | −0.1154 *** (−3.19) | −0.1008 *** (−2.18) | −0.1186 *** (−2.27) | −0.0718 *** (−2.76) | −0.0289 ** (−2.23) | −0.1009 *** (−3.07) | −0.1558 * (−1.75) | −0.2296 *** (−2.58) |
| Ln(INF) | −0.0058 (−1.42) | −0.0023 (−0.67) | −0.0069 (−1.38) | −0.0052 (−0.92) | −0.0076 (0.98) | −0.0043 (−0.71) | −0.0056 (−0.94) | −0.0190 (−1.71) |
| Ln(RE) | −0.0264 *** (−4.03) | −0.0331 *** (−4.96) | −0.0275 *** (−4.14) | −0.0342 *** (−5.13) | −0.0216 *** (−2.85) | −0.0323 *** (−4.78) | −0.0289 *** (−4.21) | −0.0227 *** (−2.76) |
| Ln(EC) | −1.0127 * (−1.65) | −1.2647 ** (−1.87) | −1.0106 (−1.59) | −1.1407 * (−1.82) | −0.5253 (−1.22) | −1.2669 ** (−2.03) | −0.9665 * (−1.59) | −0.9723 * (−1.67) |
| Spatial rho | −0.0342 *** (−3.83) | −0.0896 ** (−2.41) | −0.0352 *** (−2.87) | −0.0247 *** (−3.21) | 0.0513 (1.29) | −0.0112 (−0.21) | −0.1002 *** (−2.84) | −0.0798 ** (−2.38) |
| Spatial fixed | YES | YES | YES | YES | YES | YES | YES | YES |
| Time fixed | YES | YES | YES | YES | YES | YES | YES | YES |
| Obs | 110 | 110 | 110 | 110 | 110 | 110 | 110 | 110 |
| R$^2$ | 0.2816 | 0.3541 | 0.2807 | 0.3589 | 0.1885 | 0.3395 | 0.3259 | 0.3695 |
| Log-L | 137.5554 | 140.0668 | 136.5967 | 141.1806 | 130.8611 | 140.2660 | 142.1068 | 146.4468 |

Note: The symbols of *,**,*** in the table show that the results were significant at 90%, 95% and 99% levels, respectively. The values in brackets are standard error.

According to the results (1) and (2), the coefficients of Ln(ER1) were positive, and most of them passed the significance test of 1%. However, the regression coefficient of the quadratic term [Ln(ER1)]$^2$ was not significant when introduced in result (3), which indicates that there was a positive linear relationship between the incentive-type environmental regulation and the marine economic transformation. The reasons are that the market control methods of regional governments for marine pollution enterprises were mainly carbon emission tax and carbon emission trading; the guidance system and guarantee system related to the sustainable development of marine resources and protection of

marine environment were inadequate; under the direct influence of emission prices, micro-enterprises could make an intuitive response by improving production technology and changing energy structure.

According to the results (5) to (7), after introducing the linear term and quadratic term of investment-type environmental regulation, the coefficients of Ln(ER2) were negative, while the coefficients of quadratic term [Ln(ER2)]$^2$ were positive, and both of the coefficients passed the 1% significance test, which showed that there was a U-shaped relationship between the investment-type environmental regulation and the marine economic transformation. The reason is that under inefficient production mode, before the environmental input requirement reached an inflection point, governments and enterprises could use limited capital, which has lower environmental benefits for pollution control, which would reduce the willingness and implementation of sustainable measures such as technological innovation. In this period, the compensation effect of innovation has not been valued. The effect of investment would become more sustainable when the government and local enterprises realized that environmental income from traditional production could not make up for pollution control investment. At this stage, the roles of innovative elements, such as talent, equipment and technology, would become prominent, and the impact of environmental regulation would return to being positive. It was noteworthy that the regulation strength of China's coastal provinces is still lower than the inflection point value of 6.96, which showed that the effect of regional pollution control investment in the current stage was not positive.

According to the results (4) and (8), after adding the space-lag terms of environmental regulation to the model, the coefficients of both terms passed the significance test of 1%, which showed that both types of environmental regulations had a spillover effect on the marine economic transformation of the adjacent areas. The coefficient of W × Ln(ER1) was negative. Most of China's coastal regions were advanced areas in economic transformation and industrial restructuring, especially Shanghai and Guangdong. While strict market environmental regulations were being implemented on one side, the core regions' governments would coordinate with the adjacent areas to deduce intensity in a top-down manner. On the other side, the enterprises in core regions would amplify the pollution through cooperation with other enterprises in adjacent regions. The purpose was to transfer capacity from inefficient production to ensure the sustained growth of the whole marine economy. Therefore, while such a type of regulation would have a positive effect on the local marine economy, it would also cause the adjacent areas to become a "pollution refuge". The investment-type environmental regulation was the government's subjective behavior which could be controlled and evaluated directly by the superior government. In order to avoid the punishment from the superior government, local governments are more inclined to pursue, imitate and compete in investment with each other. The coefficient of W*Ln(ER2) was positive. It showed that local investment could reduce the pressure on pollution control in adjacent areas so that enterprises had greater transformation capacity.

The coefficients of spatial rho terms were generally negative and significant, which was consistent with the results of the spatial correlation test. The results showed that the "beggar-thy-neighbor" motivation still exists in the process of marine economic development, but the coordination of the transformation among regions needs to be improved. Although the coefficients of decentralization Ln(FD) are negative, they are not significant, which indicates that decentralization did not have a direct effect on the regional marine economy but served as an indirect moderator factor.

According to the estimation results of control variables, the regression coefficients of the level of science and technology innovation Ln(RD), financial support strength Ln(FS), marine economic contribution Ln(EC) and resource endowment level Ln(RE) passed the significance test at the 1% level, in which the impact of Ln(RD) was positive. This showed that the leading role of technological progress in the transformation of the marine economy had been highlighted. Unlike previous studies, the coefficients of Ln(FS) were negative, mainly because China's regional financial policy was more controlled by the support will of

local governments and through the selection of microeconomic subjects to intervene directly in the market. In the long-term distorted environment of the economic assessment system, local financial institutions were more willing to choose to support traditional marine industries with low short-term risk and high scale pay, resulting in technology-intensive and clean industries facing higher barriers to market access. In terms of resource endowment Ln(RE), China's marine economic development had a long-term dependence on traditional resources, such as fisheries, energy and coastal space, and formed economies of scale. The early regional resource endowment advantage had cultivated a labor-, resource-, and pollution-intensive industrial system, which was not conducive to the implantation of emerging resources under the established path dependence. This point was further validated by the negative regression coefficient of marine economic contribution. Although the influence of Ln(FDI) on an export-oriented economy was negative, it did not pass the significance test. This was different from the findings of other industries, which not only showed that the dependence of China's marine industry on external technology was still low, but also showed that the "pollution paradise" hypothesis of foreign investment in traditional industries to avoid strict environmental regulation and transfer high pollution production to backward regions has not been popularized in the field of China's marine economy.

### 4.4. The Adjustment of Decentralization on the Environmental Regulation Effect

The decentralization system of China enhances regional economic motivation and, in particular, has a profound impact on the economic strategies of adjacent regions. Therefore, this paper further explores the intersection between decentralization and environmental regulation. The results were shown in Table 8.

**Table 8.** Spatial Economic Results of the Decentralization System.

| Regulation Type | Incentive-Type | | Investment-Type | |
|---|---|---|---|---|
| Effect | Total Effect | Effect Decomposition | Total Effect | Effect Decomposition |
| Result | (9) | (10) | (11) | (12) |
| Ln(ER1) | −0.5906 *** (−2.57) | −0.5059 ** (−2.43) | | |
| [Ln(ER1)]$^2$ | 0.0009 * (0.07) | −0.0095 *** (−0.85) | | |
| W × Ln(ER1) | | −0.2032 (−4.96) | | |
| Ln(ER2) | | | 0.2371 (1.30) | 0.1964 (1.23) |
| [Ln(ER2)]$^2$ | | | 0.0274 *** (5.48) | 0.0241 *** (5.45) |
| W × Ln(ER2) | | | | 0.0907 (1.49) |
| Ln(FD) × Ln(ER1) | 0.1462 *** (2.74) | 0.1390 *** (2.90) | | |
| W × Ln(FD) × Ln(ER1) | | 0.0452 *** (5.09) | | |
| Ln(FD) × Ln(ER2) | | | −0.1219 *** (−2.94) | −0.1094 *** (−2.91) |
| W × Ln(FD) × Ln(ER2) | | | | −0.0225 (−1.63) |
| Ln(FD) | 0.4020 * (1.91) | 0.3271 * (1.72) | 0.3400 (1.55) | 0.1118 (0.53) |
| Spatial rho | −0.0256 (−0.85) | −0.2216 *** (−4.41) | 0.0895 ** (2.49) | −0.1338 *** (−2.68) |
| control | YES | YES | YES | YES |
| spatial fixed | YES | YES | YES | YES |
| time fixed | YES | YES | YES | YES |
| Obs | 110 | 110 | 110 | 110 |
| $R^2$ | 0.6932 | 0.7444 | 0.1343 | 0.3968 |
| Log-L | 95.9854 | 107.0045 | 173.9889 | 149.1284 |

Note: The symbols of *, **, *** in the table show that the results were significant at 90%, 95% and 99% levels, respectively. The values in brackets are standard error.

As shown in columns (9) and (11) of Table 7, the coefficients of both intersection terms, Ln(FD) × Ln(ER1) and Ln(FD) × Ln(ER2), were significant at the confidence level of 99%. However, the coefficient of the intersection term, which included the incentive-type environmental regulation and decentralization, was positive, showing that under the development pattern of localization, decentralization could enhance the positive effect of regional market policy on the marine economic transformation. The coefficient of intersection terms included the investment-type environmental regulation and decentralization was negative, which indicated that the decentralization system could aggravate the negative effect of pollution control investment on marine economic transformation. The results above can only reflect the comprehensive moderator effect of decentralization.

Results (10) and (12) reflect the moderator effect of decentralization on the direct and spatial spillover effects after decomposition. In the moderator effect of local decentralization, the coefficient of Ln(FD) × Ln(ER1) was positive and significant at the confidence level of 99%. This showed that the decentralization system would stimulate local governments to strengthen the market regulatory intensity and further accelerate marine economic transformation, and the positive effect also reflects that the local government has a positive attitude towards environmental benefits and prefers to strengthen the market-oriented management of the marine environment. The coefficient of Ln(FD) × Ln(ER2) was negative and significant at the confidence level of 99%. A higher decentralization level means that the local government and enterprises have more authority and ability to govern. Under the condition of investment-type environmental regulation that negatively affects marine economic transformation, the more investment, the more obvious the negative effect.

Results (10) and (12) can also reflect the moderator effect of decentralization on the spatial spillover effects of environmental regulation, in which the coefficient of W × Ln(FD) × Ln(ER1) is positive at a confidence level of 99%, showing that it could reduce the negative spillover effect of such environmental regulations. In terms of reasons, decentralization could not only further stimulate the regional governments around the core regions to strengthen marine industry spillover but also enhance the motivation to absorb advanced technology, which could ensure economic sustainability, thus reducing the transformation gap between regions caused by "pollution refuge". Although the regression coefficient of W × Ln(FD) × Ln(ER2) was negative, the moderator effect is not significant.

*4.5. Robustness Test*

The correctness of the main conclusion is closely related to whether the spatial metering weight matrix was scientific. Hence, this paper used the adjacency space weight matrix instead of the distance weight matrix to test robustness. The coefficients of the key indexes were basically the same as the empirical conclusions in this paper. It could be seen that the choice of spatial weight matrix had no substantial effect on the results, which further confirmed the reliability of the conclusions. Due to the limited space, the partial results of the robustness test were shown in Tables 9 and 10.

**Table 9.** The robustness test of environmental regulation impact on Marine Economic Transformation by changing the spatial metering weight.

| Regulation Type | Incentive-Type | | | | Investment-Type | | | |
|---|---|---|---|---|---|---|---|---|
| Result | (1) | (2) | (3) | (4) | (5) | (6) | (7) | (8) |
| Ln(ER1) | 0.0472 *** (3.25) | 0.0382 *** (2.71) | 0.0251 * (1.13) | 0.0353 *** (2.42) | | | | |
| [Ln(ER1)]² | | | 0.0022 (0.17) | −0.0012 (−0.07) | | | | |
| Ln(ER2) | | | | | 0.0162 ** (2.15) | 0.0311 *** (2.68) | −0.2472 *** (−3.76) | −0.2689 *** (−4.87) |
| [Ln(ER2)]² | | | | | | | 0.0192 *** (4.13) | 0.0243 *** (4.74) |
| W × Ln(ER1) | | −0.0213 *** −(2.37) | | −0.0238 *** (−2.55) | | | | |
| W × Ln(ER2) | | | | | | 0.0231 *** (2.62) | | 0.0196 *** (2.58) |
| Spatial fixed | YES | YES | YES | YES | YES | YES | YES | YES |
| Time fixed | YES | YES | YES | YES | YES | YES | YES | YES |
| Obs | 110 | 110 | 110 | 110 | 110 | 110 | 110 | 110 |
| R² | 0.2804 | 0.3355 | 0.2731 | 0.3472 | 0.2343 | 0.3451 | 0.3321 | 0.3704 |
| Log-L | 136.4728 | 138.2875 | 134.8447 | 140.5502 | 134.8600 | 143.5456 | 144.0997 | 149.3537 |

Note: The symbols of *, **, *** in the table show that the results were significant at 90%, 95% and 99% levels, respectively. The values in brackets are standard error. The table only shows important results.

**Table 10.** The robustness test of the moderator effect by changing the spatial metering weight.

| Regulation Type | Incentive-Type | | Investment-Type | |
|---|---|---|---|---|
| Effect | Total Effect | Effect Decomposition | Total Effect | Effect Decomposition |
| Result | (9) | (10) | (11) | (12) |
| Ln(FD) × Ln(ER1) | 0.1261 *** (2.42) | 0.1032 *** (2.51) | | |
| W × Ln(FD) × Ln(ER1) | | 0.0313 ** (3.76) | | |
| Ln(FD) × Ln(ER2) | | | −0.1031 *** (−2.72) | −0.0097 ** (−2.12) |
| W × Ln(FD) × Ln(ER2) | | | | −0.0218 (−1.57) |
| Spatial rho | −0.0256 (−0.85) | −0.2216 *** (−4.41) | 0.0895 ** (2.49) | −0.1338 *** (−2.68) |
| control | YES | YES | YES | YES |
| spatial fixed | YES | YES | YES | YES |
| time fixed | YES | YES | YES | YES |
| Obs | 110 | 110 | 110 | 110 |
| R² | 0.6642 | 0.7187 | 0.1421 | 0.4165 |
| Log-L | 93.8452 | 103.8786 | 174.5965 | 152.6541 |

Note: The symbols of **, *** in the table show that the results were significant at 95% and 99% levels, respectively. The values in brackets are standard error. The table only shows important results.

In the empirical research, the mutual causality between dependent variables and independent variables may cause endogenous problems. On the one hand, environmental regulation could affect the marine economic transformation, which has been confirmed above. On the other hand, enterprises in regions with better marine economic transformation had higher requirements for the environment, which could require local governments to increase the intensity of environmental regulation. Therefore, this paper used the two-stage least squares method of instrumental variables to test the endogeneity. The variable of environmental regulation in time $t-1$ was taken as instrumental variable. On the one hand, this variable was highly related to the current environmental regulation; on the other hand, the current marine economic transformation could not affect the previous environmental regulation. According to the results of stage 1 in Table 11, both types of environmental regulation were significantly correlated with the marine economic transformation, which indicates that there was no problem with weak instrumental variables. According to the results of stage 2, both types of environmental regulation, which were predicted by instrumental variables, had a significantly positive effect on marine economic transformation.

Therefore, the significance of instrumental variables showed that the previous studies were reliable.

**Table 11.** The robustness test by using the two-stage least squares method of instrumental variables.

| Variables | Regression of Stage 1 | | Regression of Stage 2 | |
|---|---|---|---|---|
| | ER1 | ER2 | ER1 | ER2 |
| $Ln(ER1)_t$ | | | 0.0651 *** (3.13) | |
| $Ln(ER2)_t$ | | | | 0.0112 *** (2.37) |
| $Ln(ER1)_{t-1}$ | 0.0842 *** (8.66) | | | |
| $Ln(ER2)_{t-1}$ | | 0.0510 *** (4.45) | | |
| Spatial fixed | YES | YES | YES | YES |
| Time fixed | YES | YES | YES | YES |
| Obs | 110 | 110 | 110 | 110 |
| F | 148.810 *** | 31.629 *** | | |
| $R^2$ | 0.742 | 0.480 | 0.501 | 0.468 |

Note: The symbols of *** in the table show that the results were significant at 99% level. The values in brackets are standard error. The table only shows important results.

## 5. Conclusions and Policy Implications

### 5.1. Conclusions

This paper analyzed the influence of environmental regulations on regional marine economic transformation under a decentralized system in China by applying spatial exploratory analysis and spatial econometric analysis. The data of 11 coastal provinces (or cities) from 2010 to 2019 were processed in ArcGIS 10.3 and Stata 13. The main conclusions are as follows:

First, in terms of descriptive analysis, the difference of regions' marine economic transformations had gradually narrowed. The classification distribution of the coastal provinces had gradually concentrated from high level to middle level. However, the advantages of some leading regions, such as Shanghai and Guangdong, were more obvious. The marine economic transformation and incentive-type environmental regulation showed negative spatial autocorrelation, while the correlation coefficient of investment-type regulation was significantly positive.

Second, there was a "U-shape" relationship between marine economic transformation and investment-type environmental regulation; however, the intensity in coastal regions was not enough to have a positive effect, and the effect of the innovation compensation is not obvious. The incentive-type environmental regulation had a positive linear effect on marine economic transformation. Both types of environmental regulations had spillover effects on the marine economic transformation of adjacent areas, but the effects were opposite. The spillover effect of incentive-type environmental regulation was negative, while impact of the investment-type environmental regulation was positive.

Third, the moderator effect of decentralization on incentive-type environmental regulation was positive. It could further strengthen the effect of incentive-type environmental regulation on the local economic transformation and improve the negative effect on the adjacent areas. However, the negative effect of investment-type environmental regulation on local marine economic transformation will increase along with the increased decentralization.

### 5.2. Implications

According to the evaluation results and analysis of the core variables, it can be seen that the effects of both types of environmental regulation are different, and the role of decentralization should also be improved in several aspects. There are three pieces of advice on achieving coordinated development between the marine environment and the marine economy.

First, we should improve the regional transformation willingness and establish a reasonable regional cooperation system. According to the first conclusion, the positive spatial autocorrelation of investment-type environmental regulation is mainly attributed

to the superior supervision. However, the negative spatial autocorrelation of incentive-type environmental regulation showed that regions around Shanghai and Guangdong were still willing to achieve marine economic development through pollution, which has caused a widening gap in marine economic transformation between the core regions and surrounding regions. We should build a pattern of mutual benefits and win–win results. The core regions should be a demonstration area for the marketization management of marine environments and strengthen the connection with other provinces rather than develop independently. Especially in the process of industrial cooperation, Shanghai and Guangdong should synchronously promote talent, equipment and technology transfer to surrounding regions and help marine pollution enterprises accelerate transformation. Other regional governments should realize that the goals of environmental protection and economic transformation can be achieved simultaneously. On the one hand, strengthen the supervision of enterprises' migration from core regions. On the other hand, the cost of the marine environment should be shared through payment transfers and industry synergy that could prevent unfair cooperation.

Second, we recommend that the intensity of both types of environmental regulation be strengthened and the requirements for a positive effect on marine economic transformation be met. According to the second conclusion, neither of the two types of environmental regulations has achieved their optimal effect. In terms of investment-type environmental regulation, the intensity in coastal regions was lower than the inflection point of a U-shaped relationship. Therefore, stricter policies that included more stringent discharge standards and a higher frequency of environmental protection supervision should be formulated. By strengthening these policies, the pollution cost of marine enterprises can be increased, and they will be forced to independently enhance their level of environmental protection through technological innovation and green emerging technology extension. In terms of incentive-type environmental regulation, the intensity gap between core regions and adjacent regions caused the negative spillover effect. The regional governments surrounding Shanghai and Guangdong should change the idea of "pollution havens", learn from the successful management experience from the core areas, and promote the systems of Charge for Sea Area Utilization and Emission Trading, which were dominated by market. Finally, it can increase the environmental threshold and avoid the pollution spillover caused by the migration of polluting enterprises.

Third, we should strengthen the positive role of decentralization in improving the performance of environmental regulation. According to the third conclusion, the moderator effects of decentralization on both types of environmental regulations were opposite; different strategies need to be implemented. On one hand, because of the positive moderator effect of decentralization on incentive-type environmental regulation, we should improve the enthusiasm of local governments by further empowering them to manage the marine environment in a market-oriented way. Establish the fault-tolerant mechanism for market-oriented management of the marine environment and use the information advantages of local governments to create more incentive-type environmental regulations. On the other hand, the negative moderator effect on investment-type environmental regulation showed that the greater the investment power of local governments is, the more difficult the transformation of the marine economy is. The superior government should repatriate the powers of environmental protection investment and optimize the investment structure regarding environmental protection expenditure, especially increasing the special funds for innovation and technological transformation of marine enterprises, so as to change the effect of environmental protection investment.

This paper provides proof of regional marine development uncertainty under environmental regulations and enriches the research boundary on decentralized systems. Admittedly, there are still some limitations in this paper. First, the measurement of regional environmental regulation is mainly about government investment and tax revenue, but this paper ignores other possible measurement methods, such as public participation and enterprise voluntary. In other words, the impact of new types of environmental regulation

on economic transformation is the direction to be studied. Second, this study explored whether different types of environmental regulation influenced marine economic transformation in the current year. However, it takes time for management policies to work, and there is a certain complementarity between different environmental regulations. So, the matching effect and the possibility of optimal combination at different times are a direction for subsequent research.

**Author Contributions:** H.G.: Conceptualization, visualization, formal analysis and original draft writing; C.Z.: project administration, supervision and validation; H.Z.: data curation, methodology, software and supervision; D.H.: methodology, software, writing-review and editing. All authors have read and agreed to the published version of the manuscript.

**Funding:** This research was funded by the Philosophy and Social Sciences Planning Project of Zhejiang Province in China (grant number 23NDJC255YB), Soft Science Research Program of Zhejiang Province in China (grant number 2021C35087), National Social Science Foundation of China (grant number 18VSJ023) and Public Welfare Project of Zhejiang Province (grant number LGF20G020002).

**Institutional Review Board Statement:** Not applicable.

**Informed Consent Statement:** Not applicable.

**Data Availability Statement:** The data presented in this study are available from the authors on request.

**Conflicts of Interest:** The authors declare no conflict of interest.

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
