# Peer review of "The Effect of Environmental Regulation on Marine Economic Transformation under the Decentralized System: Evidence from Coastal Provinces in China"

_sustainability, doi:10.3390/su142416622_

Round 1

Reviewer 1 Report

Authors have examined the effect of environmental regulations on marine economic transformation under the decentralized system. The topic of the paper is quite interesting, and it has significant contributions to the existing literature. I have a few suggestions to improve the quality of the paper.

1. In the introduction, authors should compare this study with previous studies, and clearly explain with arguments how this study is different from the previous studies. They should clearly describe all major contributions of this study in the introduction.

2.  In the introduction, the authors have written some research questions in bullets, I suggest writing these research questions in a paragraph.

3. One paragraph should be added at the end of the introduction to describe the sequence of the rest of the parts of the paper.

4. The authors have not shown the results of the robustness test, I suggest showing all of the estimated results

5. In table 1, the authors should add sources of data for each variable.

6. The Last section should be renamed as "Conclusions and Policy Implications". The authors should explain future recommendations in more detail. 

Author Response

Thank you very much for your opinions. We did our best to revise the manuscript according to your opinions. Please forgive us for our deficiencies.

1.According to expert opinion, the last two paragraphs of introduction explain the research objectives, innovation and major contributions.

2.According to expert opinion, the research questions were shown in one paragraph.

3.According to the expert opinion, the sequence of the rest of the parts of the paper was added in the last paragraph of the introduction.

4.According to expert opinion, Results of all robustness tests are shown.

5.According to expert advice, data sources were added in Table 1.

6.According to expert advice, the title was renamed. We rewrote the research implications that are more corresponding to the two types of environmental regulation and decentralization.

We, again, extend our sincere gratitude to you.

Reviewer 2 Report

 It is of great significance to study thethe impact of environmental regulation on the transformation of the marine economy. But the article still needs to be further improved.

 1. The core of the article is to discuss the impact of environmental regulation on the transformation of the marine economy, but the literature review lacks the literature on the impact of environmental regulation on the marine economy. In fact, there are many articles in economic and geography that discuss the impact of environmental regulation on the marine economy, and the article should increase the introduction of relevant documents.

  2. Figure 1 and Figure 2 is confusing. How does the effect mechanism of decentralization, environmental regulation, and marine economic transition construst? The construction process and evidence for this framework should be explained in detail.

3. ER 1 and ER2 impact the economy of the whole coastal province, not just the marine economy. How to distinguish their impact is on the marine economy rather than the economy of coastal provinces?、

 4. Spatial correlation analysis generally requires more than 30 research units, but 11 coastal provinces, which is surprising. It is suggested that the author change a more appropriate regression model.

 5. The research findings need further interpretation. For instance, The marine economic transformation and incentive-type environmental regulation showed negative spatial autocorrelation, while the correlation coefficient of Investment-type regulation was significantly positive. Why is this an effect? Please fully explain.

  6.Research implications are disconnected from research conclusions. The research enlightenment should be based on the research analysis, but the current research enlightenment can not be obtained without the analysis of the article.

Author Response

Thank you very much for your opinions. We did our best to revise the manuscript according to your opinions. Please forgive us for our deficiencies.

1.Based on the opinions of expert, this manuscript searched the literature on the impact of environmental regulation on Marine economy, and added a summary in Part 2.1.

2.Figure 1 shows the moderator effect of decentralization on environmental regulation. It mainly includes Strength pollution motivation by "Spillover effect", Change investment environment by "Competitive effect", Reduce pollution control Investment by "Crowding effect." Figure 2 shows the research idea of this manuscript. Based on expert advice, Figures 1 and 2 have been modified respectively and a more targeted explanation has been added.

3.This manuscript considers that although environmental regulations have a comprehensive impact on the economies of coastal provinces, the impacts on the Marine economies will also have general characteristics. In the empirical study of this manuscript, relevant data of Marine economy and Marine environment are selected to avoid estimation bias as far as possible. based on expert advice, we add relevant explanations on Index and Data Sources selection.

4.In the empirical study of this manuscript, we have made the following considerations: (1)The main objective of spatial autocorrelation analysis is to test whether a spatial econometric model is needed, so this method has to be retained in order to introduce the following text. based on expert advice, In order to increase the reliability of measurement, LM test was added before choosing the spatial measurement model, and the results also indicated that the spatial measurement model should be used. (2) This paper comprehensively analyzes the results of Spatial autocorrelation test and the results of Spatial classification. The results of both methods are consistent, which can increase the credibility of spatial autocorrelation test. (3) By referring to previous papers, it is found that spatial autocorrelation methods are also used in the study of economic correlation in China's coastal areas, although the number of regions is small. Such as: “Green innovation ability and spatial spillover effect of marine fishery in China” published in ‘Ocean & Coastal Management’, ’Spatial spillover effects of environmental regulations on air pollution: Evidence from urban agglomerations in China’ published in “Journal of Environmental Management”, ”Research on the Impact of Marine Environmental Regulation on the High-quality” published in “Journal of Ecological Economy”.

5.According to expert advice, explanations for the results of three core variables are added in Section 4.2.

6.According to expert advice, we rewrote the research implications that are more corresponding to the two types of environmental regulation and decentralization.

We, again, extend our sincere gratitude to you.

Reviewer 3 Report

This study explores the direct and spillover effects of two types of regional environmental regulations on marine economic transformation of China’s coastal provinces under a decentralized system. This paper has certain originality and innovation, but there are a few questions as follows:

1. This paper considers two type environment regulations, respectively the Incentive-type environmental regulation and the Investment-type environmental regulation. But in introduction section, there is no direct explanation of these regulations. This paper should add certain relative explanations.

2.In line 265, why does this paper introduce the lag term of the Marine Economic Transformation Index? Generally speaking, the lag term of the explained variable is introduced to build a dynamic panel model. Is the model used in this paper a dynamic panel model? And then, why the explained variable takes logarithm in the model, but the lag term of the explained variable does not take logarithm?

3. In line 277, why you choose dynamic space Durbin model? Before using Durbin model, it should pass a series of tests such as LM, Wald, LR, etc. The paper does not do any relevant tests. It is not enough to explain this problem based on the research objects, this paper introduces the space lag terms of "marine economic transformation" and "environmental regulation" and constructs the dynamic space Durbin model..

4. In Table 3. Correlation among the morphological variables do not show its significance.

5. In line 395-397, the description of According to the average values of marine economic transformation in three representative years, there was a trend of increase first and then decrease, and the change range was obviously lower than the average valueis different from the Table 4. According to Table 4, the average value shows a trend of decrease first and then increase. The description should be revised. Why do you only show these three years data? Only showing the value of marine economic transformation of 2010, 2015 and 2019 cannot explain the trend of marine economic transformation well.

6. In section 4.5, although the weight matrix is replaced for robustness test, the endogenous problem is not considered.

7. It is recommended to add some recent references.

Author Response

Thank you very much for your opinions. We did our best to revise the manuscript according to your opinions. Please forgive us for our deficiencies.

1.According to the expert opinion, We add the explanation of these environmental regulations in the third paragraph of the introduction.

2.In this paper, we consider that the "path dependence" phenomenon of economic transformation is common, and the marine economic transformation largely depends on the past level. Therefore, lag term of the Marine Economic Transformation Index is introduced in order to avoid the endogeneity caused by missing explanatory variables. This variable has actually taken logarithm, And the symbol in the formula has been modified.

3.According to the expert opinion, the relevant test results were shown, and the explanation has been added.

4.According to expert opinion, the significance symbol and the note were added in table 3.

5. According to expert opinion, the description has been modified, and the values of marine economic transformation in five years were listed in Table 4, and the purpose is to better discover the rule of change.

6.According to the expert opinion, considering that the mutual causality between dependent variable and independent variables may cause endogenous problems. we used the two-stage least squares method of instrumental variables to test the endogeneity, and take environmental regulation in time t-1 as instrumental variable. The results showed that the results of previous studies are reliable.

7.According to the expert opinions, the 12 literature of the past three years have been added, in which the relationship between environmental regulation and Marine economy has been paid more attention.

We, again, extend our sincere gratitude to you.

Round 2

Reviewer 2 Report

The author revised the article carefully. However, the policy enlightenment is relatively macro, and the policy enlightenment part can be focused and further improved, especially the research enlightenment and research conclusions should be combined.

Author Response

Thank you very much for your comment, and we cherish it very much.

According to the comment, we have read many relevant articles, and fully referred to the enlightenment of previous studies. Combine with three conclusions which include different analysis, spatial effect of environmental regulation and moderator effect of decentralization, three targeted suggestions were put forward in order to reduce the negative effects found in the paper.

If there are any other problems, we will try our best to revise them. We, again, extend our sincere gratitude to you.